# Augmentation with Projection: Towards an Effective and Efficient Data Augmentation Paradigm for Distillation

**Ziqi Wang**[1][*], **Yuexin Wu**[2][†] **Frederick Liu**[2], **Daogao Liu**[3], **Le Hou**[2], **Hongkun Yu**[2], **Jing Li**[2], **Heng Ji**[1]

[1] University of Illinois Urbana-Champaign  [2] Google  [3] University of Washington

{ziqiw9, hengji}@illinois.edu  {crickwu, frederickliu, lehou, hongkuny, jingli}@google.com
dgliu@uw.edu

## Abstract

Knowledge distillation is one of the primary methods of transferring knowledge from large to small models. However, it requires massive task-specific data, which may not be plausible in many real-world applications. Data augmentation methods such as representation interpolation, token replacement, or augmentation with models are applied to tackle this problem. However, these data augmentation methods either potentially cause shifts in decision boundaries (representation interpolation), are not expressive enough (token replacement), or introduce too much computational overhead (augmentation with models). To this end, we propose Aug-Pro (**Aug**mentation with **Pro**jection), an effective and efficient data augmentation method for distillation. Our method builds on top of representation interpolation augmentation methods to maintain the diversity of expressions and converts the augmented data to tokens to avoid shifting decision boundaries. It uses simple operations that come with little computational overhead. The results on multiple GLUE tasks show that our methods can improve distillation performance by a large margin at a low time cost. Codes are available at https://github.com/google-research/google-research/tree/master/augpro.

## 1 Introduction

Large-scale language models (Devlin et al., 2018; Raffel et al., 2020; Brown et al., 2020; Zhang et al., 2022c) have achieved great success on various natural language processing (NLP) tasks, such as information extraction (Lu et al., 2021) and question answering (Kassner & Schütze, 2020). However, large-scale models have high computational overhead, which limits their deployment in edge devices and fast response scenarios (Sun et al., 2020b). One widely used solution is to perform knowledge distillation (Hinton et al., 2015) from large-scale models to small-scale models. This method, however, usually requires a large amount of data to guarantee the transfer quality, which may not be easily obtained in real-world applications. To this end, data augmentation methods are applied (Liang et al., 2020; Wang & Yang, 2020; Zhang et al., 2022b) to improve the distillation performance.

There are three major types of data augmentation methods: (1) Representation interpolation. For example, Liang et al. (2020), Chen et al. (2020a) and Sun et al. (2020a) apply linear interpolation (Zhang et al., 2017) to word embeddings, hidden states between transformer layers, and encoder outputs, respectively, to augment the original dataset with virtual data points. Data points are virtual because they are not real language inputs. Instead, they are representations (e.g., embeddings). (2) Token replacement. Kobayashi (2018) replaces tokens with their synonyms. Easy Data augmentation (Wei & Zou, 2019) combines synonym replacement, random insertion, random swap, and random deletion. (3) Augmentation with models. Yoo et al. (2021) and Zhou et al. (2021) use GPT-3 (Brown et al., 2020) and T5 (Raffel et al., 2020) respectively as the language model to generate new text data of similar types. (1) supports many operations such as linear interpolation (Zhang et al., 2017) and

---

[*]Work was done when the first author was interning at Google.
[†]Corresponding author

small perturbation (Madry et al., 2017). It makes the methods very expressive in generating a diverse range of data. However, the newly generated representations (e.g., embeddings) may sit outside of the real data distribution. For instance, word embeddings are converted from a vocabulary in the text domain. Performing augmentation at this level may result in representations that do not have their counterparts in the vocabulary. As a result, the augmented data may mislead the model to generate a shifted decision boundary that can largely affect the qualities (Section 3). (2) can generate in-domain data easily. By using synonym replacement (Wang & Yang, 2015), new data can be obtained at a low cost. Despite this good property, this stream of methods lacks the ability to generate diversified data. Subsequently, they contribute little to sampling low-resource data areas and limit the performance gains in practice. (3) generates both diversified and in-domain data using large language models such as GPT-3 (Brown et al., 2020) and T5 (Raffel et al., 2020). Due to their large computational overheads, on the other hand, the final distillation quality will be highly limited to the amount of generated data, which is usually not affordable to the scale of even tens of thousands of sentences in practice. Figure 1 summarizes the advantages of each augmentation method.

Considering all the approaches above, we propose AugPro, an effective and efficient data augmentation method for the distillation scenario, which absorbs the advantages above without being limited by their drawbacks. Specifically, AugPro: (1) (effectiveness) is as expressive as representation interpolation; (2) (effectiveness) does not mislead decision boundaries; (3) (efficiency) has low computational overhead. In distillation settings, we can always use the teacher to label the hallucinated data in the knowledge distillation scenario. This suggests that we can encourage AugPro to produce as diverse data as possible that are not limit to instances with only the same or flipped labels.

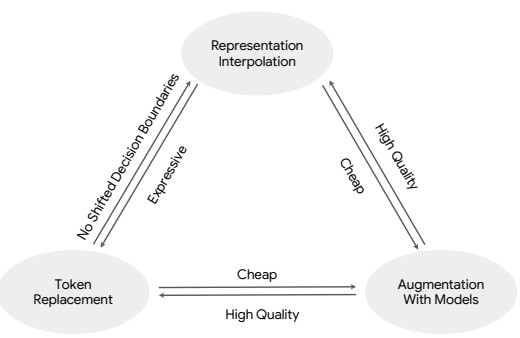

Figure 1: An illustration of each augmentation method's advantages.

Concretely, our method builds on top of representation interpolation augmentation methods (property (1)), which does not constrain the generated data to be within small regions of their "parents". The key of AugPro is to convert the augmented data to the format of tokens through projections (property (2)) with low-cost operations (property (3)). We conduct our experiments on GLUE (Wang et al., 2018) datasets. Results show that our method could boost the distillation performance significantly with low computational overhead.

To sum up, our contributions are:

- We propose an effective and efficient data augmentation method for knowledge distillation.
- We empirically evaluate the effectiveness and efficiency of AugPro and theoretically examine that AugPro satisfies three properties under certain circumstances.

## 2 RELATED WORK

**Knowledge Distillation** Knowledge distillation was first proposed by (Hinton et al., 2015). It aims to distill knowledge from one model to another by minimizing the distance between the outputs of two models on the same input. With the rise of transformers (Vaswani et al., 2017) and BERT (Devlin et al., 2018), more and more attention has been paid to the distillation of pre-training language models. Tang et al. (2019) distill fine-tuned BERT to a single-layer BiLSTM network and makes the BiLSTM network as good as ELMo (Peters et al., 2018). Sun et al. (2019) not only distill from outputs but also distill from teacher models' hidden layers. These methods distill language models in the fine-tuning stage, whereas Sanh et al. (2019) and Sun et al. (2020b) focus on distilling language models in the pre-training stage directly to make student models task-agnostic. TinyBERT (Jiao et al., 2019) distill BERT from both the pre-training and fine-tuning stages. We focus on a widely used setting. We distill knowledge in the fine-tuning stage by minimizing the distance between two models' outputs.

**Data Augmentation** Representation interpolation methods are popular in the computer vision research community. MixUp (Zhang et al., 2017) uses linear interpolation to get augmented images

and labels. Given an image $x_1$, $x_2$ and their labels $y_1$, $y_2$, MixUp uses a linear interpolation to generate a new data point $x'$ and its label $y'$:

$$x' = \text{MixUp}(x_1, x_2) = \lambda x_1 + (1 - \lambda)x_2, \quad y' = \text{MixUp}(y_1, y_2) = \lambda y_1 + (1 - \lambda)y_2 \quad (1)$$

FGSM (Goodfellow et al., 2014) and PGA (Madry et al., 2017) use gradients to generate adversarial examples. Given an image $x$, FGSM will generate new data $x'$:

$$x' = x + \epsilon \text{Sign}(\nabla_x \mathcal{L}) \quad (2)$$

where $\mathcal{L}$ is the loss of a specific task and $\epsilon$ is a small value. $x'$ and $x$ have the same label. CutMix (Yun et al., 2019) cuts images and then concatenates them together to get a new one. Though these methods were originally designed for images, they can be adapted to NLP tasks. Liang et al. (2020) use MixUp on word embeddings to augment data for knowledge distillation. Chen et al. (2020b) use MixUp on hidden states between transformer layers. Jindal et al. (2020) also use MixUp on hidden states but consider the effect of mean and variance. Zhang et al. (2022b) apply PGA to student models' embeddings and leave teacher models' embeddings unchanged, finding that PGA can benefit knowledge distillation. Token replacement methods mainly focus on language inputs. Synonym replacement methods (Kobayashi, 2018) replace tokens with their synonyms. Easy data augmentation (EDA) (Wei & Zou, 2019) incorporates synonym replacement, random insertion, random swap, and random deletion. TreeMix (Zhang et al., 2022a) uses a constituency parser to decide which token should be replaced. Augmentation with models is another approach to generating new data. FlipDA (Zhou et al., 2021) uses T5 to generate data that has flipped labels. GPT3Mix (Yoo et al., 2021) designs prompts and uses GPT3 to generate new data. Back translation (Yu et al., 2018) uses neural networks to translate inputs to another language and then translate them back. Our method (AugPro) uses representation interpolation as the backbone and utilizes projection to convert representations to tokens with low-cost operations.

## 3 MOTIVATING EXAMPLES

Though representation interpolation methods have the good property of generating diverse data, we find that these vision techniques cannot be directly applied to NLP tasks. This is because representation interpolation augments data in a continuous manner, under which case the new data may never exist in the discrete input space, causing the decision boundary shifts.

Take an example of a simple two-dimensional problem with linear separability (Figure 2). Let $\mathcal{X} = \{x_1, x_2, x_3, x_4\}$ be the universe of all data to learn, and $\mathcal{Y} = \{y_1, y_2, y_3, y_4\}$ be the corresponding labels. Suppose we know all of $\mathcal{X}, \mathcal{Y}$ and run the linear support vector machine (SVM) with a hard margin, i.e., $\min_{\beta,b} \|\beta\|^2$ such that $y_i(\beta^\top x_i + b) \geq 1$ for all $i$, we get the solution $\beta^*, b^*$ Since it is hard to get all data in the real-world setting, we suppose that we only have $\{x_1, x_3\}$ as the training dataset. If we simply use MixUp, we get $x_{\text{MixUp}}$, as the augmented data, whose label $y_{\text{MixUp}} = \text{sign}((\beta^*)^\top x_{\text{MixUp}} + b^*) = y_3 = y_4$. Now running the linear SVM with $\{x_2, x_4, x_{\text{MixUp}}\}$ with labels $\{y_2, y_4, y_{\text{MixUp}}\}$, we get $\beta_{\text{MixUp}}$ and $b_{\text{MixUp}}$. Nevertheless, $\text{sign}(\beta_{\text{MixUp}}^\top x_3 + b_{\text{MixUp}}) \neq y_3$. As a comparison, if we project $x_{\text{MixUp}}$ to its nearest neighbor and get $x_{\text{MixUp}-P} = x_2$ whose label $y_{\text{MixUp}-P} = y_2$, running SVM with $\{x_1, x_2, x_{\text{MixUp}-P}\}$ and $\{y_1, y_2, y_{\text{MixUp}-P}\}$ can get $\beta_{\text{MixUp}-P}$ and $b_{\text{MixUp}-P}$, which can classify all data correctly. Appendix B shows the concret number of each parameter.

To this end, augmented data should be the real data in the input space, i.e., the format of tokens, to leverage this problem. This observation leads to our method AugPro which uses projection to convert augmented representations to symbolic tokens. Compared with the virtual data point generated by representation interpolation, projection can explore more real data and leads to a lower error (Section 4.2 and Appendix H).

## 4 METHODOLOGY

In this section, we first formulate the definition of knowledge distillation in NLP (Hinton et al., 2015) and then introduce our method AugPro.

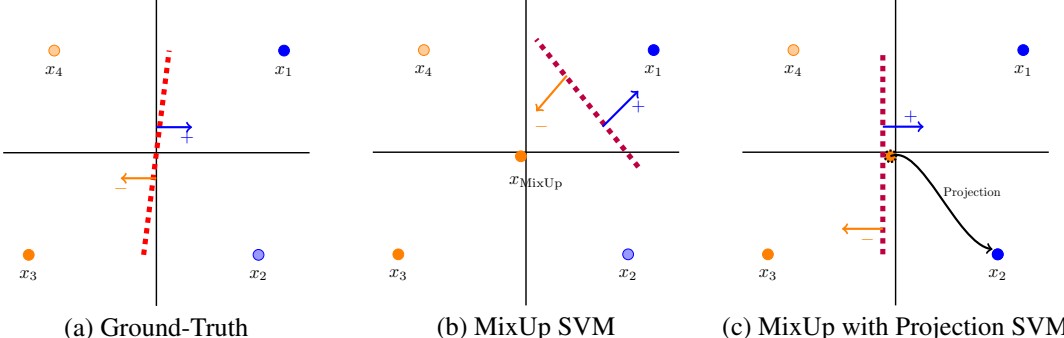

(a) Ground-Truth       (b) MixUp SVM       (c) MixUp with Projection SVM

Figure 2: Decision boundary shifts in 2D space for discrete datasets. (a) is the gound-truth, where $x_1, x_3$ are the observable training data while $x_2, x_4$ with transparent colors mean the unseen data. In (b), one gets augmented data $x_{\text{MixUp}}$ with label $y_{\text{MixUp}} = -1$, and do SVM with $\{x_1, x_3, x_{\text{MixUp}}\}$ with their labels. In (c), one projects $x_{\text{MixUp}}$ to its nearest neighbor $x_2$, and do SVM with $\{x_1, x_2, x_3\}$. We see that the correction of projection in (c) brings smaller decision boundary shifts than (b).

## 4.1 KNOWLEDGE DISTILLATION

Knowledge distillation is a method to distill knowledge from large-scale models to small-scale models. Formally speaking, considering an NLP classification task, we have a corpus $\mathcal{D} = \{(x_i, y_i)\}_{i=1}^{N}$ that contains $N$ input-output pairs, where $x_i$ is an input sentence with tokens $x_i = [w_{i_1}, \cdots, w_{i_{n_i}}], w_k \in V, V$ is the vocabulary, $n_i$ is the number of tokens in $x_i$. $y_i$ is the output label for $x_i$. We use plain texts rather than bold texts for $x$ because language inputs are a sequence of tokens, which is different from images. Then we distill knowledge from a large-scale model $f(\cdot, \theta_T)$ with parameter $\theta_T$ (i.e., a teacher model) to a small-scale model $g(\cdot, \theta_S)$ with parameter $\theta_S$ (i.e., a student model). In practice, $\theta_T$ has much more parameters than $\theta_S$. The distillation process can be divided into two stages:

- Teacher training. Optimize $\theta_T$ on the dataset $\mathcal{D}$. In classification problems, we use cross-entropy loss to do empirical risk minimization on $\theta_T$:

$$\theta_T' = \arg\min_{\theta_T} \frac{1}{N} \sum_{i=1}^{N} \text{CrossEntropy}(f(x_i, \theta_T), y_i) \tag{3}$$

- Student training. Optimize $\theta_S$ on the dataset $\mathcal{D}$ with both ground-truth labels and outputs from teachers. In classification problems,

$$\theta_S' = \arg\min_{\theta_S} \mathcal{L}_{\text{KD}} = \arg\min_{\theta_S} \frac{1}{N} \sum_{i=1}^{N} \text{CrossEntropy}(g(x_i, \theta_S), y_i) + d(g(x_i, \theta_S), f(x_i, \theta_T')) \tag{4}$$

where $d(\cdot, \cdot)$ is a distance function. In practice, $d(\cdot, \cdot)$ could be cross-entropy or mean square error.

Empirical results from former studies (Hinton et al., 2015; Sun et al., 2020b; Sanh et al., 2019; Sun et al., 2019) show that knowledge distillation will train a better $\theta_S'$ because the student model not only learns from ground-truth labels but also learns the generality from the teacher model.

Note for the student training, we can combine knowledge distillation and data augmentation together:

$$\theta_S' = \arg\min_{\theta_S} \mathcal{L}_{\text{KD}} + \mathcal{L}_{\text{Aug}}$$

where $\mathcal{L}_{\text{Aug}}$ denotes the knowledge distillation loss on augmented data which leads to different variants of methods. As one important way to help the student learn more effectively, how to generate new data with augmentation loss is the key and major discussion topic in the remaining sections.

## 4.2 AUGPRO: AUGMENTATION WITH PROJECTION

In this section, we will introduce four variants of $\mathcal{L}_{\text{Aug}}$: two backbones (MixUp and FGSM) and two AugPro variants building on top of them. Figure 4 shows the concept of our proposed method.

---

**Algorithm 1:** AugPro Algorithm

---

**Input:** Dataset $\mathcal{D} = \{(X, Y)\}$, representation interpolation function $h(\cdot)$, projection function $p(\cdot)$, the teacher model $f$ with fine-tuned parameters $\theta_T'$ and the student model $g$ with parameter $\theta_S$ that needs to be fine-tuned, vocabulary $V$, learning rate $\eta$, training steps $K$, batch size $B$, sentence length $L$, embedding dimension $H$.

**Output:** Fine-tuned $\theta_S'$

- $k = 0$
- **while** $k < K$
    - Sample a data batch $\mathcal{B} = \{(x, y)\} \in V^{B \times L}$ from $\mathcal{D} = \{(X, Y)\}$.
    - Get augmented representations $\mathcal{B}_{rep} = h(\mathcal{B}) \in \mathbb{R}^{B \times L \times H}$ (e.g., Equation (1, 2))
    - Project representations to tokens $\mathcal{B}' = p(\mathcal{B}_{rep}) \in V^{B \times L}$ (e.g., Equation 5)
    - Use Equation (4) to compute $\mathcal{L}_{\text{KD}}$.
    - Compute loss $\mathcal{L}_{\text{Aug}}$ (e.g., Equation (6, 7)) based on $\mathcal{B}'$, $f$ and $g$ .
    - $\theta_S = \theta_S - \eta \nabla(\mathcal{L}_{\text{KD}} + \mathcal{L}_{\text{Aug}}), k = k + 1$
- **return** $\theta_S$

---

AugPro builds on top of representation interpolation augmentation methods. The pipeline (Algorithm 1) can be divided into three steps: (1) We first get augmented representations (i.e. $h(\cdot)$). (2) Then we use projection to convert representations to tokens (i.e. $p(\cdot)$). (3) At last, we compute $\mathcal{L}_{\text{Aug}}$ to update student models. The key to step (2) is projection. Concretely, language models map tokens to representations, and projection aims to find an inverse mapping to map representations back to tokens. This way, AugPro will avoid shifting decision boundaries (Section 3). However, the inverse mapping is hard to find in practice. First, popular language model architectures such as transformers (Vaswani et al., 2017) are usually complex and are hard to get the inverse mapping. Second, the input space of language is discrete, making the mapping irreversible. Thus, we could only use the approximation technique to get the approximated inverse mapping. To this end, we focus on the inverse mapping on the embedding level. First, the embedding mapping's structure is much simpler than the transformer layer and is straightforward for us to find the inverse mapping. Second, we can use the nearest-neighbors as the approximation method, which is a cheap approximation.

Based on the analysis above, we use the nearest-neighbors to find our projection, i.e., the function $p(\cdot)$ in the Algorithm 1. AugPro does not rely on specific representation interpolation methods. In this paper, we apply AugPro to MixUp and FGSM, i.e. the $h(\cdot)$ in Algorithm 1 is MixUp or FGSM as in Equation (1,2). We will illustrate two variants to perform projection (step (2)) and compute $\mathcal{L}_{\text{Aug}}$ loss (step (3)) in the following texts.

We slightly abuse the notion of $f$ and $g$ to illustrate AugPro better. We divide $f$ and $g$ into two parts: the first part is an embedding function that maps tokens to embedding vectors ($f_e$ and $g_e$, $e$ denotes *embeddings*), the rest is the second part ($f_l$ and $g_l$, $l$ denotes *layers*). Under this definition, $f = f_l \circ f_e$ and $g = g_l \circ g_e$.

**AugPro-Mix** We get AugPro-Mix by applying the AugPro paradigm to MixUp. First, we apply MixUp on word embeddings and labels. This gives us embeddings from the teacher $\mathbf{e}_{\text{MixUp}}^f$, embeddings from the student $\mathbf{e}_{\text{MixUp}}^g$, and the label $y_{\text{MixUp}}$. $\mathcal{L}_{\text{Aug}}$ becomes

$$\mathcal{L}_{\text{MixUp}} = \frac{1}{M} \sum_{j=1}^{M} [\text{CrossEntropy}(g_l(\mathbf{e}_{\text{MixUp},j}^g, \theta_S), y_{\text{MixUp},j}) + d(g_l(\mathbf{e}_{\text{MixUp},j}^g, \theta_S), f_l(\mathbf{e}_{\text{MixUp},j}^f, \theta_T'))]$$

when we use MixUp data to construct losses, where $M$ denotes the number of augmented data.

For AugPro, we use the nearest-neighbors to get AugPro-Mix tokens $x_{\text{AugPro-Mix}}$:

$$x_{\text{AugPro-Mix}} = [w_{\text{AugPro-Mix},1}, \cdots, w_{\text{AugPro-Mix},n}]$$
$$\text{where } w_{\text{AugPro-Mix},i} = \max_{w \in V} \text{Sim}(\mathbf{e}_{\text{MixUp}}^f(i), \ f_e(w)) \tag{5}$$

Sim means the similarity function, which can be the cosine similarity. $\mathbf{e}(i)$ means the $i$th embedding vector in $\mathbf{e}$. Concrete examples can be found in Appendix D. Then the loss function $\mathcal{L}_{\text{Aug}}$ becomes:

$$\mathcal{L}_{\text{AugPro-Mix}} = \frac{1}{M} \sum_{j=1}^{M} d(g(x_{\text{AugPro-Mix},j}, \theta_S), f(x_{\text{AugPro-Mix},j}, \theta'_T)) \tag{6}$$

We do not use $y_{\text{MixUp}}$ because the projection operation (the nearest-neighbors in AugPro-Mix) does not necessarily preserve the label.

**AugPro-FGSM** Though adversarial examples (AE) are originally aimed to improve the robustness of models and may harm the performance on clean inputs (Raghunathan et al., 2019), Zhang et al. (2022b) shows that AE can benefit knowledge distillation. We get AugPro-FGSM by applying AugPro to FGSM. We first apply FGSM to the student model and get augmented data $\mathbf{e}^g_{\text{FGSM}}$. The augmented data can be used to construct $\mathcal{L}_{\text{Aug}}$ directly:

$$\mathcal{L}_{\text{FGSM}} = \frac{1}{M} \sum_{j=1}^{M} d(g_l(\mathbf{e}^g_{\text{FGSM},j}, \theta_S), f_l(\mathbf{e}^f_j, \theta'_T))$$

Following Equation 5, we could get $x_{\text{AugPro-FGSM}}$ by changing the footnotes accordingly.

We usually set $\epsilon$ in Equation (2) large in AugPro-FGSM since we do not want $x_{\text{AugPro-FGSM}}$ to be the same as the original input, whereas $\epsilon$ in Equation (2) is a small value in FGSM. We use the cosine similarity to implement the Sim function. The loss function $\mathcal{L}_{\text{Aug}}$ becomes:

$$\mathcal{L}_{\text{AugPro-FGSM}} = \frac{1}{M} \sum_{j=1}^{M} d(g(x_{\text{AugPro-FGSM},j}, \theta_S), f(x_{\text{AugPro-FGSM},j}, \theta'_T)) \tag{7}$$

**Label Diversity** We take two sentences from SST-2 dataset (Socher et al., 2013) as an example to further explain that projection does not necessarily preserve labels but generates diverse labels. The first sentence is `watch on video at home` with the sentiment `Neutral`. The second sentence is `as good` with the sentiment `Positive`. Then we can get the AugPro-Mix sentence `watch good video at home`. Obviously the label of AugPro-Mix sentence should be `Positive` rather than the linear interpolation of `Positive` and `Neutral`. This is the desired property in distillation as we will use the teacher to label these newly generated data points.

**Computational Overhead**. If we assume the complexity of computing cosine similarity between two vectors is $\mathcal{O}(d)$ where $d$ is the dimension of vectors, then the complexity of the projection (the nearest-neighbors in our implementation) is $\mathcal{O}(NVd)$, where $N$ is the sentence length and $V$ is the vocabulary size. $N$ is usually within hundreds. $V$ is usually around $30,000$ in popular pre-train language models using sub-word tokens such as BERT (Devlin et al., 2018) and T5 (Raffel et al., 2020). As a result, $\mathcal{O}(NVd)$ brings little costs. On the other hand, the projection operation could be parallelized since the $NV$ similarity calculations do not affect each other. In modern parallel computing architectures, such as GPUs and TPUs, projection can be calculated in a much faster manner. Compared to the major large-scale language models' complexities, this computation will take a small portion of resources. The detailed running time comparison can be found in Section 5.2.

**Three Properties of AugPro.** Since AugPro supports operations used in representation interpolation methods, AugPro is expressive (property (1)). AugPro also converts representations to tokens to avoid shifting decision boundaries, leading to a smaller error rate (property (2)). It can be shown that AugPro-Mix has a $\frac{1}{4N}$ lower error rate than MixUp, and AugPro-FGSM has a $\frac{1}{2N}$ lower error rate than FGSM with certain assumptions (Appendix H). Moreover, AugPro has a low computational overhead to guarantee efficiency (property (3)), as described in the previous paragraph.

## 5 EXPERIMENTS

Our experiments aim to answer two questions: (1) How effective is AugPro when applied to the knowledge distillation scenario? (2) Is AugPro efficient?

**Datasets and Settings** Following previous knowledge distillation works (Liang et al., 2020; Zhang et al., 2022b), we use GLUE (Wang et al., 2018) datasets as the benchmark. We use EncT5 (Liu

et al., 2021) as our teacher and student models for the following reaons: (1) T5 has a much better performance than BERT and is close to SOTA in many tasks. EncT5 is a simplified version of T5 that uses the whole encoders of T5 but only one decoder layer. EncT5 performs similarly to T5 on classification tasks such as GLUE tasks with fewer parameters. For example, EncT5 (small) only contains 37M parameters but can perform similarly to T5 (small), which contains 77M parameters. Using EncT5 will make the results more convincing and show that our method is still useful even with powerful models. (2) Previous methods (Liang et al., 2020; Zhang et al., 2022b) distill knowledge from a 12-layer BERT to a 6-layer BERT or a 3-layer BERT. However, the gap between the teacher and student models is marginal. Therefore, the improvement space is limited, and the existence of variance will weaken the credibility of the results. To this end, we distill knowledge from EncT5 (Large, 24-layer, 354M, teacher) to EncT5 (small, 8-layer, 37M, student), as the two models have a significant performance gap.

**Baselines and Training** We train several baselines for comparison: (1) **Fine-Tuning (FT)**: We directly fine-tune EncT5 on the dataset. (2) **Knowledge Distillation (KD)**: We first fine-tune a teacher model (EncT5 Large), then distill knowledge from the teacher model to the student model (EncT5 Small). (3) **Knowledge Distillation + Back Translation (KD+BT)**: Back translation (Yu et al., 2018) first translates input to another language and then translates it back. We choose back translation as a representative method for the data augmentation type "augmentation with models". (4) **Knowledge Distillation + K-Nearest-Neighbors (KD+KNN)** KNN (Wang & Yang, 2015) first selects tokens from inputs, then replaces them with the K nearest neighbors in the embedding space. KNN can be regarded as one token replacement method. (5) **KD+MixUp** (6) **KD+FGSM** (7) **KD+TMix** (Chen et al., 2020b) MixUp on the hidden state between transformer layers. The last three methods are of the "representation interpolation" type. We train student models with 0.6M steps and 512 batch size. Due to the high computation cost, we only augment data to twice as large as the original dataset size for back translation. For all other methods, we augment data to twice as large as the original batch size for each batch, i.e., we augment $0.6M$ steps $\cdot$ $512$ batch size $= 307.2M$ data in total. More training details are in Appendix C.

## 5.1 EFFECTIVENESS OF AUGPRO

Table 1 shows the results of knowledge distillation. Due to the high cost, we only report back translation results on the RTE dataset. We first use the training data to train a teacher model and then distill knowledge from the teacher model to the student model on the training data. We can conclude that: (1) All data augmentation methods will benefit the distillation. (2) AugPro can significantly improve the distillation performance compared with corresponding baselines. Specifically, AugPro is extremely useful for low-resource datasets such as CoLA and RTE. AugPro-Mix achieves scores 5.97% and 9.02% higher than MixUp on CoLA and RTE, respectively. AugPro-FGSM achieves scores 10.52% and 8.31% higher than FGSM on CoLA and RTE, respectively. For large datasets such as MNLI, AugPro-Mix and AugPro-FGSM can also improve the performance. (3) Moreover, combining AugPro-FGSM and AugPro-Mix achieves the best performance in all listed methods. Compared with vanilla knowledge distillation, combining AugPro-Mix and AugPro-FGSM improves the performance from 2% to 14%.

Table 2 uses a different setting from Table 1. We only keep 10% training data labeled and assume others are unlabeled. Then we use labeled training data to train a teacher model and unlabeled training data to do knowledge distillation—this is a more realistic setting since it is often easier to get unlabeled data than to get labeled data. The conclusions above still hold. Specifically, AugPro can improve the accuracy from 1% to 2% on average on three datasets. Compared with the vanilla distillation, AugPro can improve around 2% accuracy at most on three datasets.

## 5.2 EFFICIENCY OF AUGPRO

The efficiency of AugPro lies in two aspects. First, its complexity is low. Second, it can be computed in parallel. To fully demonstrate these two advantages, we report the real-time cost of AugPro and baselines in Table 3. KD+data augmentation is rough twice the time of vanilla KD since these methods use twice the data as vanilla KD. We can also observe that augmentation with models (KD+BT) takes much more time than other kinds of baselines, which shows that this method is not efficient enough. At last, AugPro brings little computational overhead as the time cost is the same as

| | SST-2
Acc
67.3k | CoLA
Matthew
8.5k | MNLI-MM/M
Acc
392.7k | QNLI
Acc
104.7k | QQP
Acc/F1
363.8k | MRPC
Acc/F1
3.7k | STS-B
PC/SC
5.7k | RTE
Acc
2.5k |
|---|---|---|---|---|---|---|---|---|
| EncT5$_{24}$-FT (354M) | 97.20 | 63.60 | 91.40/91.10 | 95.40 | 92.73/90.00 | 91.42/93.30 | 88.19/88.00 | 86.30 |
| EncT5$_8$-FT   (37M) | 92.89 | 45.84 | 84.69/84.26 | 89.84 | 91.45/88.41 | 87.01/90.91 | 86.39/85.94 | 59.21 |
| EncT5$_8$-KD | 92.09 | 45.56 | 85.93/85.61 | 89.46 | 91.36/88.24 | 84.56/88.85 | 87.29/87.18 | 61.37 |
| +BT | - | - | - | - | - | - | - | 61.73 |
| +KNN | 94.27 | 54.60 | 87.13/87.01 | 91.54 | 92.14/89.40 | 86.03/90.32 | 87.14/87.27 | 66.79 |
| +TMix | 93.35 | 44.42 | 86.79/86.79 | 91.10 | 91.76/88.84 | 87.25/90.97 | 87.52/87.35 | 63.18 |
| +MixUp | 93.23 | 51.63 | 86.73/86.69 | 91.31 | 91.82/88.97 | 88.48/91.68 | 87.47/87.33 | 62.82 |
| +AugPro-Mix | 94.38 | 57.60 | 87.40/87.27 | 92.06 | 92.06/89.23 | 89.46/92.34 | 88.10/87.87 | 71.84 |
| +FGSM | 92.20 | 46.37 | 85.88/85.53 | 89.58 | 91.21/88.06 | 84.56/89.23 | 87.56/87.26 | 62.09 |
| +AugPro-FGSM | 94.61 | 56.89 | 87.02/86.85 | 91.67 | 92.10/89.26 | 88.24/91.67 | 87.64/87.51 | 70.40 |
| +FGSM+MixUp | 93.12 | 50.79 | 86.85/86.65 | 91.09 | 91.75/88.85 | 87.25/90.88 | 87.15/87.00 | 62.82 |
| +AugPro-FGSM+AugPro-Mix | **95.18** | **59.01** | **87.97/87.87** | **92.92** | **92.30/89.54** | **89.46/92.42** | **88.34/88.04** | **74.73** |

Table 1: Knowledge distillation on the GLUE dataset. We first use the training data to train a teacher model and then distill knowledge from the teacher model to the student model on the training data. EncT5$_L$ denotes EncT5 with L transformer layers. L = 24 and L = 8 denote the teacher model with 354M parameters and the student model with 37M parameters, respectively.

| | KD | +MixUp | +AugPro-Mix | +FGSM | +AugPro-FGSM | +FGSM+MixUp | +AugPro-Mix+AugPro-FGSM |
|---|---|---|---|---|---|---|---|
| MNLI-MM/M | 84.81/84.39 | 85.53/85.33 | 86.36/85.87 | 85.07/84.58 | 85.87/85.82 | 85.70/85.65 | **86.76/86.81** |
| SST-2 | 92.09 | 93.35 | 94.04 | 92.09 | **94.27** | 93.46 | 94.04 |
| QNLI | 89.68 | 90.28 | 90.98 | 89.99 | 91.03 | 90.50 | **91.58** |

Table 2: Knowledge distillation on the GLUE dataset with a different setting from Table 1. We regard 10% of the data as labeled and the rest as unlabeled. The teacher model is first trained on labeled training data and then used for knowledge distillation on unlabeled training data.

the baselines. Results also show that KNN is much slower than other methods, which is explained in Appendix G.

## 5.3 ABLATION STUDY

In ablation studies, we follow settings used in Table 1 unless otherwise stated.

**Perturbation scale for $\epsilon$ in AugPro-FGSM** The key hyperparameter in AugPro-FGSM is $\epsilon$ in Equation (2). Small $\epsilon$ will make $x_{\text{AugPro-FGSM}}$ the same as the original input. Large $\epsilon$ tends to make $x_{\text{AugPro-FGSM}}$ hard to understand, meaningless, and out of the domain. Therefore, a proper $\epsilon$ is essential. Our experiments find that $\epsilon = 35$ is the best fit for T5 embeddings. Table 4 shows KD+AugPro-FGSM performance with different $\epsilon$.

**Signs of gradients in AugPro-FGSM are not important** The effectiveness of AugPro-FGSM comes from gradients' signs and the projection in AugPro. To prove that AugPro-FGSM mainly benefits from AugPro, we implement two AugPro-FGSM variants: AugPro-FGSMD (Descent Projection) that uses the opposite signs to AugPro-FGSM, and AugPro-FGSMR (Random Projection) that uses random signs. Table 6 shows the results of AugPro-FGSM and its two variants. We can observe AugPro-FGSM has a similar score to its variants in all settings. Thus AugPro-FGSM mainly benefits from AugPro. We also conduct experiments that follow the setting of Table 2, and results can be found in Appendix E.

**AugPro generates diverse labels** We show that AugPro generates diverse labels at the end of Section 4. Here we empirically show that assuming AugPro preserving labels may harm performance. If AugPro preserves labels, AugPro-Mix and AugPro-FGSM data should have the same labels as MixUp and original data, respectively. We use these augmented data together with labels to fine-tune student models directly. Results in Table 5 suggest that such augmented data and labels may harm performance. Therefore, AugPro generates diverse labels and does not necessarily preserve labels.

**AugPro consistently benefits KD with different data sizes** Figure 3 shows the performance of AugPro with different data sizes. It can be observed that AugPro is better than all baselines in all data sizes. Moreover, AugPro is extremely useful when the data size is small. For example, AugPro can

---
[1]Converted time. We run BT on TPU v2 and compute the equivalent time cost.

|  | KD | +BT[1] | +KNN | +TMix | +MixUp | +AugPro-Mix | +FGSM | +AugPro-FGSM |
|---|---|---|---|---|---|---|---|---|
| Time (min) | 1.68 | 13.15 | 4.57 | 3.48 | 3.48 | 3.48 | 3.24 | 3.24 |

Table 3: Time (minutes) costs every 1000 steps on average of various methods on 8 TPU v3 slices. Costs contain data augmentation, the forward pass, and the backpropagation. The table is divided into four parts. Each part contains a specific data augmentation type (KD, augmentation with models, token replacement, representation interpolation and AugPro).

| $\epsilon$ | MNLI-MM/M | SST-2 | QNLI |
|---|---|---|---|
| **30** | 85.60/85.28 | 93.69 | 90.44 |
| **35** | **85.87/85.82** | **94.27** | **91.03** |
| **40** | 85.79/85.58 | 93.23 | 90.81 |
| **100** | 85.18/84.74 | 91.97 | 90.44 |

Table 4: KD+AugPro-FGSM performance with different $\epsilon$.

| Data Size | 20% | 50% | 100% |
|---|---|---|---|
| **Finetune** | **80.99/80.71** | **83.43/82.55** | **84.69/84.26** |
| **AugPro-Mix** | 80.62/80.44 | 83.24/82.60 | 84.53/83.92 |
| **AugPro-FGSM** | 80.74/80.32 | 83.16/82.44 | 84.37/83.61 |

Table 5: AugPro-Mix and AugPro-FGSM are used to fine-tune student models. MixUp labels are used for AugPro-Mix data. AugPro-FGSM uses the original label.

| KD | MNLI-MM/M | SST-2 |
|---|---|---|
| **+AugPro-FGSM** | 87.02/86.85 | **94.61** |
| **+AugPro-FGSMD** | **87.08/86.89** | 94.27 |
| **+AugPro-FGSMR** | 86.57/86.52 | 94.27 |
| **+MixUp+AugPro-FGSM** | 87.35/87.41 | **94.50** |
| **+MixUp+AugPro-FGSMD** | 87.45/87.44 | 94.04 |
| **+MixUp+AugPro-FGSMR** | **87.48/87.49** | 94.15 |
| **+AugPro-Mix+AugPro-FGSM** | **87.97/87.87** | **95.18** |
| **+AugPro-Mix+AugPro-FGSMD** | 87.81/87.67 | 94.61 |
| **+AugPro-Mix+AugPro-FGSMR** | 87.77/87.62 | 94.95 |

Table 6: KD+AugPro-FGSM and its variants performance with different signs. AugPro-FGSMD denotes FGSM with Decent Projection. AugPro-FGSMD uses the opposite sign to AugPro-FGSM . AugPro-FGSMR denotes FGSM with Random Projection. AugPro-FGSMR uses the random sign.

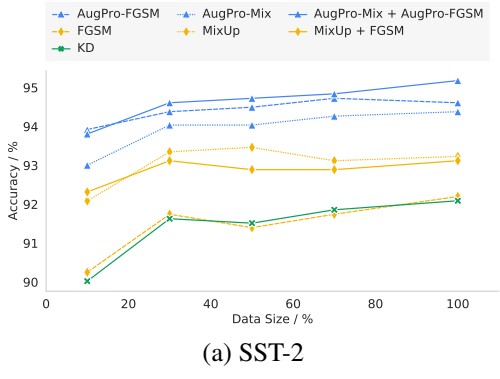

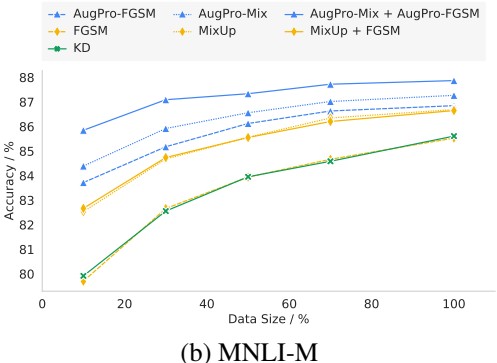

(a) SST-2      (b) MNLI-M

Figure 3: AugPro performance with different data sizes. Figure (a) and Figure (b) are for SST-2 dataset and MNLI-M dataset. Blue lines (or triangle markers) are AugPro methods. Yellow lines (or diamond markers) are baseline methods. The green line (or X marker) is KD. AugPro has the same line type as the corresponding baseline. For example, AugPro-FGSM and FGSM are all dashed lines.

improve the accuracy of 4% (SST-2) and 6% (MNLI-M) when the data size is 10%. We also report results on the MNLI-MM dataset in Appendix F.

## 6 CONCLUSIONS AND FUTURE WORK

We propose AugPro, an effective and efficient data augmentation paradigm for knowledge distillation. We use projections to tackle the problem of shifting decision boundaries caused by traditional representation interpolation methods in knowledge distillation. Moreover, AugPro has low computation costs and is fast in modern computing architectures. Results on GLUE tasks prove the effectiveness and efficiency of AugPro. In the future, we will further explore the impact of AugPro on labels to make it helpful in other scenarios.

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

## A    THE CONCEPT FIGURE OF AUGPRO

Here we show a concept figure (Figure 4) to let readers better understand the difference between AugPro (e.g., AugPro-Mix) and previous works (e.g., MixUp (Zhang et al., 2017)).

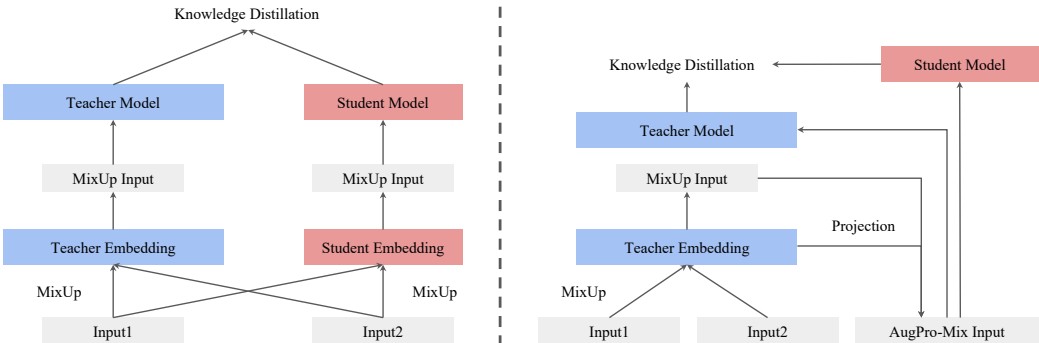

Figure 4: **Left**: MixUp with knowledge distillation. **Right**: AugPro-Mix with knowledge distillation.

## B    CONCRETE NUMBERS OF THE EXAMPLE IN SECTION 3

$\mathcal{X} = \{x_1 = (2.5, 2), x_2 = (2, -2), x_3 = (-2.5, -2), x_4 = (-2, 2)\}$
$\mathcal{Y} = \{y_1 = +1, y_2 = +1, y_3 = -1, y_4 = -1\}$

- $\beta^* = (4/9, -1/18), b^* = 0$.

- $x_{\text{MixUp}} = \frac{13}{25}x_3 + \frac{12}{25}x_1, \beta_{\text{MixUp}} = (250/533, 200/533), b_{\text{MixUp}} = -12/13$

- $\beta_{\text{MixUp}-P} = (4/9, 0), b_{\text{MixUp}-P} = 1/9$

## C    IMPLEMENTATION DETAILS

### C.1    HYPERPARAMETERS

We use JAX and T5X to implement EncT5 and AugPro, and use T5 1.1 checkpoints to initialize models. The batch size is 512, and the maximum sentence length is 128. Training steps are 0.6M for all experiments. We use 8 TPU v3 slices to do all experiments.

To fine-tune the teacher model, we use a dropout rate of 0.1 and a learning rate of 1e-3 for all GLUE tasks.

For knowledge distillation, we set the dropout rate to be 0.1 for both the teacher and student models. We find that adding dropout to teachers will make the distillation better. We run all experiments with 1e-3 and 1e-5 learning rates and report the best results. As a result, learning rate is set to 1e-5 for all experiments on the STSB dataset, **EncT5$_8$-FT** and **EncT5$_8$-KD** experiments on CoLA, MRPC and RTE datasets. All other experiments use the learning rate 1e-3.

The $\lambda$ (Equation (1)) for AugPro-Mix  is 0.5. Previous works (Zhang et al., 2017; Liang et al., 2020) use a beta distribution to sample $\lambda$ for MixUp. We try $\lambda \sim Beta(0.4, 0.4)$ and $\lambda = 0.5$ for MixUp and find they have similar performance in most tasks. In some tasks such as CoLA, $\lambda = 0.5$ is better. Therefore, we use $\lambda = 0.5$ for MixUp.

Following previous works (Zhou et al., 2021), we set $k = 15$ for the KNN baseline and randomly select 0.1 portion of tokens to replace. We use outputs of the 4th layer of the student model and the 12th layer of the teacher model, i.e., the middle layer of both models, to conduct TMix experiments.

### C.2 Implementation detail of Equation 5

Algorithm 1 shows the AugPro pipeline. Here, we show a detailed implementation of Equation 5 in Algorithm 2. We believe these two algorithms can help readers reproduce our methods and results.

---

**Algorithm 2:** Projection Algorithm

---

**Input:** Augmented representations $\mathcal{B}_{rep} \in \mathbb{R}^{B \times L \times H}$, where $B$ is the batch size, $L$ is the sentence length, and $H$ is the embedding dimension. Vocabulary $V$ contains $Z$ tokens. Token embeddings $\mathbf{E} \in \mathbb{R}^{Z \times H}$.

**Output:** Projected augmented data $\mathcal{B}' \in V^{B \times L}$

- Compute the similarity between each token pair $\mathbf{Sims} = \mathcal{B}_{rep} \cdot \mathbf{E}^T \in \mathbb{R}^{B \times L \times Z}$

- Find the nearest neighbor $\mathcal{B}'_{index} = \underset{axis=2}{\arg\min} \; \mathbf{Sims} \in \mathbb{N}^{B \times L}$

- Find tokens according to indices to get $\mathcal{B}'$

- **return** $\mathcal{B}'$

---

## D Examples of AugPro-Mix and AugPro-FGSM

We use three MNLI inputs to show the augmented data by AugPro-Mix and AugPro-FGSM.

Following are three MNLI inputs:

(i) **hypothesis:** `The judgments need to consider the broader public interest.`
**premise:** `These judgments need to be made in a consistent manner with consideration of the broader public interest in the program or activity under review.`

(ii) **hypothesis:** `I agree.`
**premise:** `yeah i i agree with that`

(iii) **hypothesis:** `She's never been to a hospital.`
**premise:** `and uh as if that wasn't bad enough the the ones that were half alive that they rushed to the hospital and she got to work on and she got to see them die and uh and they just get all the world's worst situations very few rewarding situations uh and`

If we use AugPro-Mix on (i) and (ii), (ii) and (iii), we will get:

(i + ii) **hypothesis:** `I judgment.`
**premise:** `yeah judgmenti needi agree made in`

(ii + iii) **hypothesis:** `She agree. never been to a hospital.`
**premise:** `yeah uh as agreeif that wasn't bad enough the the ones that were half alive that they rushed to the hospital and she got to work on and she got to see them die and uh and they just get all the world's worst situations very few rewarding situations uh an`

If we use AugPro-FGSM on (i), (ii) and (iii), we will get:

(i) **hypothesis:** `River judgment Mit handy to consider the broader public interest.`
**premise:** `These judgment Mit handy to be made in a consistent manner with consideration– the broader public interest in the federal clădire activity under reviewed.`

(ii) **hypothesis:** `Kyle agree.`
**premise:** `yeah chambres chambres agree with that`

| | SST-2 Acc 67.3k | CoLA Matthew 8.5k | MNLI-MM/M Acc 392.7k | QNLI Acc 104.7k | QQP Acc/F1 363.8k | MRPC Acc/F1 3.7k | STS-B PC/SC 5.7k | RTE Acc 2.5k |
|---|---|---|---|---|---|---|---|---|
| EncT5$_{24}$-FT (354M) | 97.20 | 63.60 | 91.40/91.10 | 95.40 | 92.73/90.00 | 91.42/93.30 | 88.19/88.00 | 86.30 |
| EncT5$_8$-FT (37M) | 92.89 | 45.84 | 84.69/84.26 | 89.84 | 91.45/88.41 | 87.01/90.91 | 86.39/85.94 | 59.21 |
| EncT5$_8$-KD | 92.09 | 45.56 | 85.93/85.61 | 89.46 | 91.36/88.24 | 84.56/88.85 | 87.29/87.18 | 61.37 |
| +Random | 92.78 | 45.89 | 86.37/85.55 | 89.35 | 91.42/88.43 | 84.07/88.93 | 87.72/87.57 | 61.37 |
| +MixUp | 93.23 | 51.63 | 86.73/86.69 | 91.31 | 91.82/88.97 | 88.48/91.68 | 87.47/87.33 | 62.82 |
| +AugPro-Mix | 94.38 | 57.60 | 87.40/87.27 | 92.06 | 92.06/89.23 | 89.46/92.34 | 88.10/87.87 | 71.84 |

Table 7: Results of the random generation baseline.

(iii) **hypothesis:** She' Mit never been to a hospital.
**premise:** and uh as if that wasn' financing bad enough the the ones that are half alive that they rushed to the hospital and she got to work on and she got to see them die and uh and they just get all the world' Mit worst situations very few rewarding situations uh and

The above-augmented sentences are originally token lists, but not real sentences. Luckily, T5 uses SentencePiece to construct its vocabulary, which supports the precise de-tokenization for any token list. To convert token lists to sentences, we use a simple command `"".join(tokens).replace("_"," ")`.

**Linguistic Analysis.** Our motivation focuses on the perspective of machine learning, i.e., avoiding shifting decision boundaries by converting representations to tokens. Here, we would like to add a brief analysis from a linguistic perspective. We can observe that the above-augmented sentences may have grammatical errors, "meaningless" tokens, and may be less meaningful than original sentences. However, "meaningless" to humans does not suggest meaningless to models, as AugPro indeed boosts the distillation performance. Besides, augmented sentences are not totally semantically meaningless to humans. It is hard to see why augmented data is so helpful from the linguistic perspective, suggesting that we should focus on analyzing these data from the machine learning perspective, which is exactly our motivation. To further support our motivation, we conduct a simple baseline "random generation" that randomly chooses tokens from the vocabulary and concatenates them to form an augmented sentence. Random generation can generate meaningless sentences easily. Results are shown in Table 7. We can conclude that random generation is a poor augmentation method, suggesting that AugPro is helpful not because of the meaningful or meaningless semantics, but because of avoiding shifting decision boundaries.

# E SIGNS OF GRADIENTS IN AUGPRO-FGSM IS NOT IMPORTANT (MORE RESULTS)

Table 8 shows the effect of signs of gradients in AugPro-FGSM with the setting of Table 2. The conclusion is same as the conslusion concluded from Table 6.

# F AUGPRO CONSISTENTLY BENEFITS KD WITH DIFFERENT DATA SIZES (MORE RESULTS)

The main texts show that AugPro consistently benefits KD with different data sizes on SST-2 and MNLI-M datasets. Figure 5 shows this conclusion still holds on the MNLI-MM dataset. The overall trend is similar to the MNLI-M dataset.

# G KNN IS SLOWER THAN OTHER METHODS

There are two reasons that KNN is slower than other methods. First, KNN has a higher computational complexity $\mathcal{O}(NVd\log k)$ than AugPro ($\mathcal{O}(NVd)$). Second, KNN is hard to be implemented by XLA. A popular and fast implementation of KNN is to use `np.argpartition`. However, XLA

|  | MNLI-MM/M | SST-2 | QNLI |
|---|---|---|---|
| **KD+AugPro-FGSM** | **85.87**/**85.82** | **94.27** | 91.03 |
| **KD+AugPro-FGSMD** | **85.87**/85.52 | 93.92 | **91.05** |
| **KD+AugPro-FGSMR** | 85.6/85.39 | 93.35 | 90.85 |
| **KD+MixUp+AugPro-FGSM** | 86.20/86.28 | 93.81 | 90.99 |
| **KD+MixUp+AugPro-FGSMD** | **86.44**/86.19 | 93.92 | 90.98 |
| **KD+MixUp+AugPro-FGSMR** | 86.38/**86.42** | 94.04 | 91.34 |
| **KD+AugPro-Mix+AugPro-FGSM** | **86.76**/**86.81** | 94.04 | 91.58 |
| **KD+AugPro-Mix+AugPro-FGSMD** | 86.59/86.57 | 93.81 | 91.56 |
| **KD+AugPro-Mix+AugPro-FGSMR** | 86.62/86.48 | 93.69 | 91.4 |

Table 8: KD+AugPro-FGSM and its variants performance with different signs. AugPro-FGSMD denotes FGSM with Decent Projection. AugPro-FGSMD uses the opposite sign to AugPro-FGSM . AugPro-FGSMR denotes FGSM with Random Projection. AugPro-FGSMR uses the random sign. This table follows the setting in Table 2.

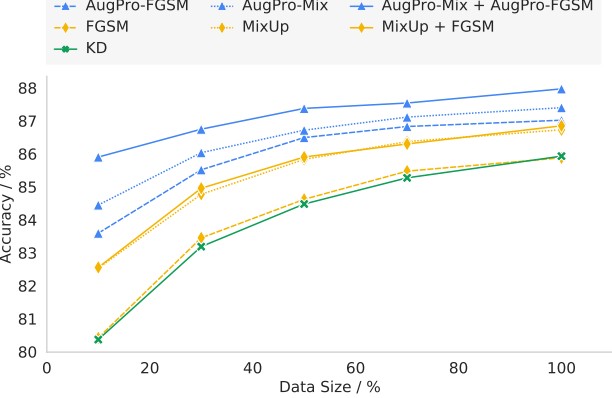

Figure 5: AugPro performance with different data sizes on MNLI-MM dataset. Blue lines (or triangle markers) are AugPro methods. Yellow lines (or diamond markers) are baseline methods. The green line (or X marker) is KD. AugPro has the same line type as the corresponding baseline. For example, AugPro-FGSM and FGSM are all dashed lines.

does not support `partition` operations[2], making KNN hard to be implemented with JAX on TPUs. To this end, Chern et al. (2022) propose another method to run KNN on TPUs at peak FLOP/s. We use their method to implement KNN, but such implementation may not be the optimal solution for KNN.

## H    AUGPRO CAN ACHIEVE LESS ERROR RATES

Suppose $\mathcal{X} = \{-1, 1\}^{2 \log n}$ is the universe of all data with labels $\mathcal{Y}$ to learn. Define a distribution $\mathcal{P}$ such that $(x, y) \sim \mathcal{P}$ if $x$ is uniformly independently drawn from $\mathcal{X}$ and $y$ is uiformly independently drawn from $\{-1, +1\}$. $\mathcal{X}_\mathcal{D}$ is the training data compositing of $n$ samples uniformly and independently drawn from $\mathcal{X}$.

Suppose $n$ is even. Let $x(j)$ be the $j$-th coordinate of vector $x$. We construct $\mathcal{X}_{\text{AugPro-Mix}}$ as follows: Index the elements in $\mathcal{X}_\mathcal{D}$ arbitrarily, and $x_i$ denotes the $i$-th element. For each positive integer $i \in [n/2]$, we construct an augmented data $z_i$ such that, for each coordinate $j \in [2 \log n]$, $z_i(j) = x_{2i-1}(j)$ if $x_{2i-1}(j) = x_{2i}(j)$, otherwise $z_i(j)$ is uniformly chosen from $\{1, -1\}$.

Let's consider optimal algorithms $\mathcal{A}$ and $\mathcal{A}_{\text{AugPro-Mix}}$. $\mathcal{A}$ can only observe $\mathcal{X}_\mathcal{D}$ (or $\mathcal{X}_{\text{MixUp}}$, $\mathcal{X}_{\text{FGSM}}$) and their labels $\mathcal{Y}_\mathcal{D}$ (or $\mathcal{Y}_{\text{MixUp}}$, $\mathcal{Y}_{\text{FGSM}}$). For proof, we assume $\mathcal{A}$ can only observe $\mathcal{X}_\mathcal{D}$ and their labels $\mathcal{Y}_\mathcal{D}$. The other two can be proved similarly. $\mathcal{A}_{\text{AugPro-Mix}}$ can observe $\mathcal{X}_{\text{AugPro-Mix}}$ and their corresponding labels $\mathcal{Y}_{\text{AugPro-Mix}}$.

Define the generalization error $\text{error}(\mathcal{A}) := \Pr_{(x,y)\sim\mathcal{P},(\mathcal{X}_\mathcal{D},\mathcal{Y}_\mathcal{D})}[\mathcal{A}(x, \mathcal{X}_\mathcal{D}, \mathcal{Y}_\mathcal{D}) \neq y]$, where $\mathcal{A}(x, \mathcal{X}_\mathcal{D}, \mathcal{Y}_\mathcal{D}) \in \{+1, -1\}$ is the prediction of $x$ outputted by $\mathcal{A}$, and $\text{error}(\mathcal{A}_{\text{AugPro-Mix}}) = \Pr_{(x,y)\sim\mathcal{P},(\mathcal{X}_{\text{AugPro-Mix}},\mathcal{Y}_{\text{AugPro-Mix}})}[\mathcal{A}(x, \mathcal{X}_{\text{AugPro-Mix}}, \mathcal{Y}_{\text{AugPro-Mix}}) \neq y]$. We have the following claim:

**Claim 1.** *One has*

$$\lim_{n\to\infty} \frac{\text{error}(\mathcal{A}) - \text{error}(\mathcal{A}_{\text{AugPro-Mix}}))}{1/(4n)} = 1.$$

As for constructing $\mathcal{X}_{\text{AugPro-FGSM}}$, for each $x_i \in \mathcal{X}_\mathcal{D}$, we add Gaussian $\mathcal{N}(0, 4)$ to each coordinate of $x_i$, and then project back to $\mathcal{X}$ to get $w_i$. Similarly, assume $\mathcal{A}_{\text{AugPro-FGSM}}$ can observe the dataset $\mathcal{X}_{\text{AugPro-FGSM}}$ and its labels $\mathcal{Y}_{\text{AugPro-FGSM}}$, and define the generalization error to be $\text{error}(\mathcal{A}_{\text{AugPro-FGSM}}) = \Pr_{(x,y)\sim\mathcal{P},(\mathcal{X}_{\text{AugPro-FGSM}},\mathcal{Y}_{\text{AugPro-FGSM}})}[\mathcal{A}(x, \mathcal{X}_{\text{AugPro-FGSM}}, \mathcal{Y}_{\text{AugPro-FGSM}}) \neq y]$.

**Claim 2.** *One has*

$$\lim_{n\to\infty} \frac{\text{error}(\mathcal{A}) - \text{error}(\mathcal{A}_{\text{AugPro-FGSM}})}{1/(2n)} = 1.$$

### H.1    PROOF OF CLAIM 1

We provide the following preliminary background of martingale to prove Claim 1 for completeness.

**Definition 1** (Martingale). *A sequence of random variables $Y_1, Y_2, \cdots$ is a martingale with respect to another sequence $X_1, X_2, \cdots$ if for all $n$, $\mathbb{E}[|Y_n|] < \infty$ and $\mathbb{E}[Y_{n+1} \mid X_1, \cdots, X_n] = Y_n$.*

**Definition 2** (Martingale Difference). *$\{D_k\}_{k=1}^\infty$ is a martingale difference sequence w.r.t. $\{X_k\}_{k=1}^\infty$ if for all $n$:*

- *$D_n$ is a measurable function of $X_1, \cdots, X_n$*

- *$\mathbb{E}[|D_n|] < \infty$*

- *$\mathbb{E}[D_{n+1} \mid X_1, \cdots, X_n] = 0.$*

If $Y_k$ is a martingale, then $D_k = Y_k - Y_{k-1}$ is a martingale difference. For martingale difference, we have the following theorem:

---

[2]https://github.com/google/jax/issues/10541

**Proposition 1** (Freedman's Inequality, Freedman (1975))**.** *Consider a real-valued martingale difference sequence $\{X_t\}$ which is uniformly bounded, i.e. $|X_t| \leq M$ almost surely for all $t$. Define the predictable quadratic variation process of the martingale $W_t := \sum_{j=1}^{t} \mathbb{E}[X_j^2 \mid \mathcal{F}_{j-1}]$ for all $t$, where $\{\mathcal{F}_t\}$ is the filtration. Then for all $\ell \geq 0$ and $\sigma^2 > 0$,*

$$\Pr\left[\exists k \geq 0 : |\sum_{t=1}^{k} X_t| \geq \ell \ \& \ W_k \leq \sigma^2\right] \leq 2\exp\left\{\frac{-\ell^2/2}{\sigma^2 + M\ell/3}\right\}.$$

With these tools, we are ready to prove the claim.

*Proof of Claim 1.* In our setting, it is evident that $\text{error}(\mathcal{A}) = \Pr_{x \sim \mathcal{P}}[x \notin \mathcal{X}_\mathcal{D}]/2 = 1/2 - \mathbb{E}[|\mathcal{X}_\mathcal{D}|]/2n^2$, where the expectation is taken over the randomness of $\mathcal{X}_\mathcal{D}$, and $|\cdot|$ denotes the cardinality after removing duplicate elements.

Similarly, we have $\text{error}(\mathcal{A}_{\text{AugPro-Mix}}) = 1/2 - \mathbb{E}[|\mathcal{X}_{\text{AugPro-Mix}}|]/2n^2$. It suffices to prove

$$\lim_{n \to \infty} \frac{\mathbb{E}[|\mathcal{X}_{\text{AugPro-Mix}}| - |\mathcal{X}_\mathcal{D}|]}{n/2} = 1.$$

First, we show for some constant $c_1 > 0$,

$$\Pr\left[|\mathcal{X}_\mathcal{D}| \geq n - c_1\sqrt{n\log n}\right] \geq 1 - 1/\text{poly}(n). \tag{8}$$

It is equivalent to drawing the elements in $\mathcal{X}_\mathcal{D}$ one by one. Let $\mathcal{X}_\mathcal{D}^i$ denote the elements after drawing $x_i$. Let $Y_i = |\mathcal{X}_\mathcal{D}^i|$, and $D_i$ be the indicator that $x_i \neq x_j$ for all $0 < j < i$, where $D_1 = 1$. Then we know $|\mathcal{X}_\mathcal{D}| = Y_n$, $D_i = Y_i - Y_{i-1}$, and for any $\mathcal{X}_\mathcal{D}^{i-1}$,

$$\Pr[D_i = 1 \mid \mathcal{X}_\mathcal{D}^{i-1}] = 1 - \frac{|\mathcal{X}_\mathcal{D}^{i-1}|}{|\mathcal{X}|} \geq 1 - \frac{1}{n}.$$

Let $\mathcal{F}_i$ be the filtration, and hence $\mathbb{E}[D_i \mid \mathcal{F}_{i-1}] \geq 1 - \frac{1}{n}$. Let $\tilde{D}_i := D_i - \mathbb{E}[D_i \mid \mathcal{F}_{i-1}]$ be a martingale difference sequence. Hence $|\tilde{D}_i| \leq 1$ almost surely, and $\mathbb{E}[\tilde{D}_i^2 \mid \mathcal{F}_{i-1}] \leq 1$. Let $W_i = \sum_{j=1}^{i} \mathbb{E}[\tilde{D}_j^2 \mid \mathcal{F}_{j-1}]$. By Freedman's Inequality, one has

$$\Pr[Y_n < n - c_1\sqrt{n\log n}]$$
$$= \Pr[\sum_{i=1}^{n} D_i < n - c_1\sqrt{n\log n}]$$
$$\leq \Pr[|\sum_{i=1}^{n} \tilde{D}_i| \leq c_1\sqrt{n\log n}]$$
$$= \Pr[|\sum_{i=1}^{n} \tilde{D}_i| \leq c_1\sqrt{n\log n} \wedge W_n \leq n]$$
$$\leq 2\exp(\frac{-c_1^2 n\log n}{2n + 2c_1\sqrt{n\log n}/3}).$$

Choosing $c_1$ large enough proves Equation (8).

Similarly we show for some constant $c_2 > 0$,

$$\Pr\left[|\mathcal{X}_{\text{AugPro-Mix}}| \geq 3n/2 - 3c_2\sqrt{n\log n}\right] \geq 1 - 1/\text{poly}(n). \tag{9}$$

It is equivalent to constructing $\mathcal{X}_{\text{AugPro-Mix}}$ by iterations. For $i$-th iteration, we draw $x_{2i-1}$ and $x_{2i}$ i.i.d. uniformly and construct $z_i$ as described before. Let $\mathcal{X}_{\text{AugPro-Mix}}^i$ denote the data we get after constructing $x_{2i-1}, x_{2i}$ and $z_i$. Let $Y_i' = |\mathcal{X}_{\text{AugPro-Mix}}^i|$ and $D_i'$ be the indicator that $Y_i' - Y_{i-1}' = 3$ (i.e. all of the three data are distinct and first constructed). We know $Y_{n/2}' \geq 3\sum_{i=1}^{n/2} D_i'$. For any

fixed vector $x \in \{-1, 1\}^{2 \log n}$, we know $\Pr[x_{2i-1} = x] = \Pr[x_{2i} = x] = \Pr[z_i = x] = 1/n^2$. Hence for any $\mathcal{X}^{i-1}_{\text{AugPro-Mix}}$ and by union bound, one has

$$\mathbb{E}[D'_i = 1 \mid \mathcal{X}^{i-1}_{\text{AugPro-Mix}}] \geq 1 - \frac{3(|\mathcal{X}^{i-1}_{\text{AugPro-Mix}}| + 3)}{|\mathcal{X}|} \geq 1 - \frac{9}{2n}.$$

Let $\mathcal{F}'_i$ be the filtration and hence $\mathbb{E}[D'_i \mid \mathcal{F}'_{i-1}] \geq 1 - \frac{9}{2n}$. Let $\tilde{D}'_i := D'_i - \mathbb{E}[D'_i \mid \mathcal{F}_{i-1}]$ be a martingale difference sequence. Similarly $|\tilde{D}'_i| \leq 1$ almost surely and $\mathbb{E}[\tilde{D}'^2_i \mid \mathcal{F}_{i-1}] \leq 1$. Let $W'_i = \sum_{j=1}^{i} \mathbb{E}[\tilde{D}'^2_j \mid \mathcal{F}'_{j-1}]$. By Freedman's Inequality, one has

$$\Pr[Y'_{n/2} < 3n/2 - 3c_2\sqrt{n \log n}]$$
$$\leq \Pr[\sum_{i=1}^{n/2} D'_i < n/2 - c_2\sqrt{n \log n}]$$
$$\leq \Pr[|\sum_{i=1}^{n/2} \tilde{D}'_i| \leq c_2\sqrt{n \log n} - O(1)]$$
$$= \Pr[|\sum_{i=1}^{n/2} \tilde{D}'_i| \leq c_2\sqrt{n \log n} - O(1) \wedge W'_{n/2} \leq n/2]$$
$$\leq 2\exp(\frac{-c'^2_2 n \log n}{n + 2c'_2\sqrt{n \log n}/3}),$$

where $c'_2 = c_2 - O(1)$. Choosing $c_2$ large enough completes the proof of Equation (9).

Combining equations (8) and (9) proves the statement. □

## H.2 PROOF OF CLAIM 2

*Proof.* Similarly, we have $\text{error}(\mathcal{A}_{\text{AugPro-FGSM}}) = 1/2 - \mathbb{E}[|\mathcal{X}_{\text{AugPro-FGSM}}|]/2n^2$ and it suffices to prove

$$\lim_{n \to \infty} \frac{\mathbb{E}[|\mathcal{X}_{\text{AugPro-FGSM}}| - \mathbb{E}[|\mathcal{X}_{\mathcal{D}}|]]}{n} = 1.$$

We already have Equation (8). It suffices to prove that for some constant $c_3 > 0$,

$$\Pr\left[|\mathcal{X}_{\text{AugPro-FGSM}}| \geq 2n - c_3\sqrt{n \log n}\right] \geq 1 - 1/\text{poly}(n). \tag{10}$$

For each $x_i \in \mathcal{X}_{\text{AugPro-FGSM}}$ and each coordinate $j \in [2 \log n]$, we know $\Pr[x_i(j) = w_i(j)] = \Phi(1/2) < 0.7$, where $\Phi$ is cumulative distribution function (CDF) of one-dimensional standard Gaussian distribution, i.e. $\Phi(t) = \Pr_{x \sim \mathcal{N}(0,1)}[x \leq t]$.

It is equivalent to constructing $\mathcal{X}_{\text{AugPro-FGSM}}$ iteration by iteration, where in $i$-th iteration, we draw $x_i$ and construct $w_i$ as described before. Let $\mathcal{X}^i_{\text{AugPro-FGSM}}$ be the data we get after constructing $x_i$ and $w_i$, let $Y_i = |\mathcal{X}^i_{\text{AugPro-FGSM}}|$ and let $D_i$ be the indicator that $Y_i - Y_{i-1} = 2$. For any fixed vector $x \in \{-1, 1\}^{2 \log n}$, we know $\Pr[x_i = x] = 1/n^2 = \Pr[w_i = x]$ and $\Pr[w_i = x_i] < 0.7^{2 \log n} < 1/n^{1.02}$. Let $\mathcal{F}_i$ be the filtration, and for any $\mathcal{X}^{i-1}_{\text{AugPro-FGSM}}$, by union bound, one has

$$\mathbb{E}[D_i = 1 \mid \mathcal{X}^{-1}_{\text{AugPro-FGSM}}] \geq 1 - \frac{2|\mathcal{X}^{i-1}_{\text{AugPro-FGSM}}|}{|\mathcal{X}|} - 1/n^{1.02} \geq 1 - 3/n.$$

Let $\tilde{D}_i := D_i - \mathbb{E}[D_i \mid \mathcal{F}_{i-1}]$ be a martingale difference sequence. We know $|\tilde{D}_i| \leq 1$ and $\mathbb{E}[\tilde{D}_i \mid \mathcal{F}_{i-1}] \leq 1$ almost surely. Let $W_i = \sum_{j=1}^{i} \mathbb{E}[\tilde{D}_i^2 \mid \mathcal{F}_{j-1}]$. By Freedman's Inequality, one has

$$\Pr[Y_n < 2n - 2c_3\sqrt{n \log n}]$$
$$\leq \Pr[\sum_{i=1}^{n} D_i < n - c_3\sqrt{n \log n}]$$
$$\leq \Pr[|\sum_{i=1}^{n} \tilde{D}_i| < c_3'\sqrt{n \log n} \wedge W_n \leq n]$$
$$\leq 2\exp(\frac{-c_3'^2 n \log n}{2n + 2c_3'\sqrt{n \log n}/3}),$$

where $c_3' = c_3 - O(1)$. Choosing $c_3$ large enough completes the proof.

Combing Equation (8) and (10) completes the proof. $\qquad\square$

