# OpenReview forum: "Augmentation with Projection: Towards an Effective and Efficient Data Augmentation Paradigm for Distillation"
_ICLR.cc/2023/Conference — ICLR 2023 poster_

### Official Review · Reviewer_55Yt · 2022-10-19

**Confidence:** 2
**Correctness:** 3
**Technical Novelty And Significance:** 2
**Empirical Novelty And Significance:** 2
**Recommendation:** 6

**Clarity, Quality, Novelty And Reproducibility:**

I think the overall writing and motivation clarity, quality need to be improved, yet, they are not critical and could be improved during the rebuttal. Also, the method is quite novel but I am not sure the proposed method contains important technical contributions.

**Strength And Weaknesses:**

- Strength
  - The method is simple yet effective and address limitations of existing three types of data augmentation methods.
  - On the GLUE benchmark consisting of 8 downstream tasks, this method shows consistent results outperforming the baselines.

- Weaknesses

[Major] The major concern is that the motivation is unclear for the reviewer. Could the authors provide more detailed explanations about the below questions?

1) In section 3, I am confused about the relation between decision boundary shifts problem and domain difference property between CV and NLP. I think motivation example in (b) of Figure 2 can occur in CV domain too, but the first paragraph in Section 3, the authors said that the problem is that representation interpolation in CV domain can not directly applied to NLP tasks. I think labels in vision classification tasks are also discrete, mixup  makes interpolation between two discrete labels. Why discrete space in NLP tasks (Positive / Negative) is a problems when we try to existing data augmemtation methods?

2) Following the (b) and (c) in Figure 2, for the projection, it looks that we should use the unseen data (x2). Could you explain more in terms of the fair comparison and detailed description about actual unseen data used in the proposed method?

3) Why do we need to combine the proposed AugPro to existing data augmentation methods? Could we use the proposed AugPro solely?

4) If I read correctly, this method said that choosing the existing tokens is better than generating new tokens based on interpolation. Could the authors provide the detailed interpretation about this?

[Minor] The writing need to be improved. I think that the contributions can be more well explained in the introduction.
[Minor] Where is the results validated on WNLI which is one of datasets in GLUE benchmark?


**Summary Of The Paper:**

This paper proposes a novel and efficient data augmentation method for knowledge distillation, by building on top of representation interpolation augmentation methods. Existing data augmentation methods can be divided as three types and have their own limitations as follows: 1) representation interpolation - the newly generated samples might be out-of-distribution 2) token replacement - lack of ability to generate diversified data 3) augmentation with models - large computational overhead. To address such problems, in this work, the proposed method aims to find inverse mapping to map representations back to tokens on the embedding level. Especially, they use nearest-neighbors as the approximated inverse function. With little computational overhead, the proposed method improves the distillation performance on multiple tasks in GLUE benchmark.

**Summary Of The Review:**

This paper proposes a simple yet effective data augmentation method for KD on NLP domain overcoming the limitations of existing methods. Also, they empirically validated the performance of the method on GLUE benchmark, which is one of main benchmark in NLP domain. However, there are some unclear parts in the motivation and why this method works well. Thus, during the rebuttal period, I hope the authors address the questions that I raised.

---

> ### Author Response · Authors · 2022-11-16
> **Response to reviewer 55Yt**
>
> Thank you for your careful review and valuable feedback! We are happy that you find our method useful.
>
> ### W1: In section 3, I am confused about the relation between decision boundary shifts problem and domain difference property between CV and NLP. I think motivation example in (b) of Figure 2 can occur in CV domain too, but the first paragraph in Section 3, the authors said that the problem is that representation interpolation in CV domain can not directly applied to NLP tasks. I think labels in vision classification tasks are also discrete, mixup makes interpolation between two discrete labels. Why discrete space in NLP tasks (Positive / Negative) is a problems when we try to existing data augmentation methods?
> A: We agree with the reviewer that both CV and NLP have discrete labels. However, we focus more on inputs as they are much more important than labels for distillation (teacher models can generate labels). Inputs of CV (i.e., images) are different from inputs of NLP (i.e., sentences). Images can be treated as “continuous” inputs because they are [W, H, C] matrices with float numbers. Sentences, on the contrary, are “discrete” inputs because they are sequences of tokens. As a result, MixUp can be directly applied to images but not sentences. One solution is to conduct MixUp token-wise on the embedding layer. Such a solution may shift decision boundaries because augmented embeddings may not have corresponding tokens and never appear in the actual input spaces, which is different from images.
>
>
> ### W2: Following the (b) and (c) in Figure 2, for the projection, it looks that we should use the unseen data ($x\_2$). Could you explain more in terms of the fair comparison and detailed description about actual unseen data used in the proposed method?
>
> A: We apologize for causing you confusion. We indeed use $x\_2$ as augmented data in real implementation. In our algorithm, $x\_1, x\_2, x\_3, x\_4$ are the whole vocabulary (T5 vocabulary). $x\_1$ is the $i$th token’s embedding of sentence 1, and $x\_3$ is the $i$th token’s embedding of sentence 2. $x\_{MixUp}$ is the token embedding of the linear interpolation that does not exist in the real vocabulary. We project $x\_{MixUp}$ to $x_2$ by finding the nearest neighbor of $x\_{MixUp}$ in the whole vocabulary. Appendix C shows examples of actual generated data.
> The projection will not drive a generated example too far from the distribution because it is in the actual input space ($x\_{MixUp}$ is not because it does not have a corresponding token in the vocabulary). Thus, $x_2$ is better than $x\_{MixUp}$ in the actual application, which is shown in our main results (AugPro-Mix vs. MixUp).
>
> ### W3: Why do we need to combine the proposed AugPro to existing data augmentation methods? Could we use the proposed AugPro solely?
> A: The reviewer raises an interesting question about using AugPro solely. The key of AugPro is to convert augmented representations to tokens, namely projection. The whole augmentation process is:  original inputs -> augmented representations (e.g. new token embeddings) -> augmented tokens. The second arrow is AugPro, and the first can be any representation interpolation methods, such as MixUp and FGSM. Thus, a complete pipeline should contain AugPro and an arbitrary representation interpolation method.
>
>
> ### W4: If I read correctly, this method said that choosing the existing tokens is better than generating new tokens based on interpolation. Could the authors provide the detailed interpretation about this?
> A: We are sorry for causing you such confusion. The linear interpolation does not generate real tokens. For example, if token A has an embedding [0.1 0.2 0.3], and token B has an embedding [0.4 0.5 0.6], then linear interpolation will generate a new embedding [0.25 0.35 0.45] (weight = 0.5). However, such an embedding may not have a corresponding token in the vocabulary. Thus, linear interpolation can only generate “virtual” tokens. AugPro, on the contrary, can generate a real token by finding the token that has the closest embedding to [0.25 0.35 0.45] in the whole vocabulary. This sometimes will be “existing” training tokens but in general, any data points in predefined vocabulary will help to “anchor” the embedding to a better place in the vocabulary space.
>
> As the whole input space is all possible tokens, “virtual” tokens are not in the input space, causing a shifted decision boundary. However, AugPro generates data in the input space, which alleviates this problem. We hope our response can solve your question.

---

> > ### Author Response · Authors · 2022-11-16
> > **Response to reviewer 55Yt (Cont.)**
> >
> > ### W5: [Minor] The writing need to be improved. I think that the contributions can be more well explained in the introduction
> > A: Please see our general response.
> >
> >
> > ### W6: [Minor] Results on WNLI dataset
> > A: Thanks for pointing out this missing dataset! Here we show the results that all use the same random seed. Results still support our claims. Our methods (AugPro-*) can still obtain the best results.
> >
> > | EncT5$\_{24}$-FT |  EncT5$\_{8}$-FT | EncT5$\_{8}$-KD |  +MixUp | + AugPro-Mix | +FGSM | +AugPro-FGSM | +MixUp+FGSM | +AugPro-Mix+AugPro-FGSM |
> > |-|-|-|-|-----|-|-----|-|-|
> > | 81.69                |           45.07     |         40.85       |    43.66   |     **56.34**          |  50.70    |      **56.34**             |       47.89           |     **56.34**     |
> >
> >
> > We originally did not report WNLI results because it is hard to get stable results as the dataset is too small (635 training data and 71 dev data). Random seeds can easily and highly affect results, making themselves less convincing. Nevertheless, we believe that results on the other 8 GLUE tasks can support our claims well.

---

> > > ### Comment · Reviewer_55Yt · 2022-11-27
> > > **Thank the authors for the response.**
> > >
> > > After carefully reading authors' response, I think the authors addressed my concerns, so I stand my score (6).

---

### Official Review · Reviewer_ZbR2 · 2022-10-24

**Confidence:** 3
**Correctness:** 3
**Technical Novelty And Significance:** 3
**Empirical Novelty And Significance:** 3
**Recommendation:** 8

**Clarity, Quality, Novelty And Reproducibility:**

The idea is novel and the results are encouraging. It is supported with also theoretical analysis. It is already in the good form to publish.

**Strength And Weaknesses:**

Strengths:

1. The approach is sensible and novel to me in term of projecting to mapping the embedding to tokens.

2. The results are encouraging and outperform the baseline augmentations on GLUE tasks.

3. The paper is very well-written and easy to follows.

Weakness:

1. It is unclear to me why the overhead is small as I have to search nearest token for each iteration. Perhaps because it leverages the parallel computation in TPU? It would be good if the authors can detail a bit more how the inverse mapping is implemented?

2. Can the authors explain why AutoProj is applied suitable for distillation? As it seems to me that the method is quite general and can applied to training the teacher network alone?

**Summary Of The Paper:**

The paper addresses data augmentation for distillation one NLP task. Existing interpolation representation method has the issues that shifts the detection boundary in NLP because NLP space exists with discrete inputs. The proposed method addresses the limitations of interpolation representation to maintain the diversity of expression and avoid the shifting decision boundary. The main idea is trying to augment the real data in input space. In particular, it augments data to get augmented representation (feature embedding at middle layers) and project the augmented representation into tokens by the inverse mapping into nearest real data in the input space. Using these nearest real data to compute the augmentation loss. The projection (called AutoProj) is the main contribution in this work. Two main augmentations (MixUp and FGSM) are considered in the work and AutoProj is built on top of it. This method is simple with little computation overhead but improve the distillation performance with a large margin. Experiments on language models show the encouraging results compared to other data augmentation methods.

**Summary Of The Review:**

Overall, the paper is well-written and the idea is interesting. All claims are well-supported with good results and ablation study.

---

> ### Author Response · Authors · 2022-11-16
> **Reponse to reviewer ZbR2**
>
> Thank you for your detailed review! We are glad that you find our paper novel and easy to follow.
>
> ### W1: It is unclear to me why the overhead is small as I have to search for the nearest token for each iteration. Perhaps because it leverages the parallel computation in TPU? It would be good if the authors can detail a bit more how the inverse mapping is implemented?
>
> A: The inverse mapping is the nearest-neighbor algorithm, and the cosine similarity defines the distance. Explicitly speaking, let us assume $\mathbf{x}\_{rep}$ is one instance of $\mathcal{B}\_{rep}$, and $\mathbf{x}\_{rep} = [\mathbf{h}\_1, \cdots, \mathbf{h}\_n]$, where $\mathbf{h}\_i$ is the augmented embedding $\in \mathbb{R}^d$ and $d$ is the embedding dimension. The vocabulary $V$ has $|V|$ tokens, and each word $w\_i \in V$ has a corresponding embedding $\mathbf{e}\_i \in \mathbb{R}^d$. Then the pseudo-code of Equation 5 is as follows:
>
> \begin{equation}
>   \begin{aligned}
>      for\ \mathbf{h}\_i\ in&\ \mathcal{x}\_{rep}: \\\\
>     for\ &w\_j\ in\ V: \\\\
>     	&similarity\_{ij} = cosine(\mathbf{e}\_j, \mathbf{h}\_i) \\\\
>     w'\_{i} =  &argmax\_{w\_j}\ similarity_{ij}
>   \end{aligned}
> \end{equation}
>
> Where $w'\_{i}$ is a token $\in V$ and the output of AugPro for $\mathbf{h}\_i$.
>
> This algorithm is much faster than other algorithms (e.g., using T5 and GPT-3 to do augmentation). In practice, as Table 3 shows, it consumes unnoticeable time. This can be explained by the following two reasons (also briefly mentioned in the last two paragraphs in Section 4):
>
> * The size of $V$ is small. Previous work [1] finds that going through the whole vocabulary is time-consuming and not practical because traditional vocabularies (e.g., word2vec, glove) usually have billions of tokens. However, with the sub-word technique, modern transformer architectures usually only have ~30,000 tokens (e.g., BERT, T5), a considerable deduction from the vocabulary size. A small size of $V$ makes this algorithm very fast.
> * We can observe that both for-loops in the above algorithm can be computed in parallel because previous iterations do not affect the computation of the next iteration. For example, a simple command ```#pragma omp parallel for``` can be added before the for-loop to achieve this goal. In practice, these two for-loops can be simplified to simple matrix multiplication and be computed paralleled on  TPUs or GPUs to speed up the computation.
>
>
> One round of the algorithm will be in $O(|V| n d + |V|n)$ where $|V|$ is the vocabulary size, and $n$ is the sequence length (and MixUp will be $O(n d)$ where d is the word embedding dimension << $|V|$). This new cost will be much smaller than the general training time for the transformer, where the multi-head attention will take $O(2n^2 d+3nd^2)$, and the FFN part will take $O(2nmd)$ (2-layer MLP), where $m$ is the MLP dimension. Concretely, student models have $8$ encoder layers, and teacher models have $24$ encoder layers. $|V| = 32000, n = 128, d=512$ and $1024, m= 1024$ and $2816$ for student and teacher models, respectively. It is not hard to compute that one inference time $8 \times (2n^2 d_s +3nd_s^2+ 2nm_sd_s) + 24 \times (2n^2 d_t +3nd_t^2+ 2nm_td_t)$ is seven to eight times more than the AugPro time $|V|nd_t+ |V|n$, where the footmark $s$ denotes the student model and $t$ denotes the teacher model. In reality, this number would be higher considering the existence of decoder layers (T5 has the same number of decoder layers as encoder layers, EncT5 only has 1 decoder layer), normalization operators, and activation functions. Besides, backpropagation time is four to five times more than the inference time. Moreover, AugPro can be computed fully paralleled, whereas model inference cannot (we have to get the previous layer’s output before conducting inference on the current layer). All these factors make the AugPro cost negligible in real runtime with the help of parallel computing architectures (e.g., TPUs and GPUs).
>
> We apologize for the confusion and have illustrated our implementation of above pseudo-codes in Appendix B.2 in the updated PDF.
>
>
> > [1] Yong Cheng, Lu Jiang, and Wolfgang Macherey. 2019. Robust Neural Machine Translation with Doubly Adversarial Inputs. In Proceedings of the 57th Annual Meeting of the Association for Computational Linguistics, pages 4324–4333, Florence, Italy. Association for Computational Linguistics.

---

> > ### Author Response · Authors · 2022-11-16
> > **Response to reviewer ZbR2 (Cont.)**
> >
> > ### W2: Can the authors explain why AutoPro is applied suitable for distillation? As it seems to me that the method is quite general and can applied to training the teacher network alone?
> >
> > A: Thank you for pointing out a more general scenario! As Section 4 discusses, AugPro can generate data with diverse labels. In other words, it is hard to determine labels of augmented data. Thus, these data are unlabeled data and are not suitable for training teacher networks. However, the such property does not affect distillation as teacher networks can generate soft labels for student networks. Moreover, AugPro data is not restricted to a small area of the original data point as it does not need to be restricted by preserving labels like other methods (e.g., EDA, synonym replacement), making it a better choice for the distillation scenario. If we find the labels of AugPro data, we can apply AugPro to a more general scenario. We thank the reviewer for pointing out this promising direction; we will leave this as our future work.

---

### Official Review · Reviewer_FUtZ · 2022-10-25

**Confidence:** 2
**Correctness:** 4
**Technical Novelty And Significance:** 3
**Empirical Novelty And Significance:** 3
**Recommendation:** 6

**Clarity, Quality, Novelty And Reproducibility:**

Clarity: the method is pretty clear.

Quality: improvements seem to be consistent across tasks.

Novelty: AFAIK the method is novel, but not extraordinarily so.

Reproducibility: I don't believe the code is available. The authors added implementation details in Appendix B, but that's probably not sufficient to reproduce the experiments.

**Strength And Weaknesses:**

Strengths:
- A concrete and well-scoped problem with a reasonable solution
- consistently beats baselines

Weaknesses:
- I'm not sure how much the final sentences make sense as we'll be adding token i's from sentence 1 with token i's from sentence 2. That can totally destroy any sort of meaning. Then again, in knowledge distillation, we might not care about this and only care about matching two functions point-by-point. I'm not sure how that's better than creating gibberish sentences as augmentation though.


**Summary Of The Paper:**

The authors propose a novel method for data augmentation to be used for knowledge distillation in NLP. It has been shown that heavy augmentation is highly useful for knowledge distillation (in vision tasks), but the same methods are not applicable to NLP due to the tokens being discrete. Therefore, previous works tend to ignore the first layer and apply the interpolation at the representation layer.
The authors use a simple trick in order to get around this issue: they use NN matching over the vocabulary as their approximate inverse projection, which allows them to mix the representations at the embedding layer, but still create valid tokens/input samples.

**Summary Of The Review:**

The problem is well-defined, and the solution is simple and to the point. The improvements also seem to be consistent across the board. The paper is also well-written and easy to follow with multiple examples. I am not an expert in this field, so I can't be confident, but I believe this paper meets the threshold for acceptance.

---

> ### Author Response · Authors · 2022-11-16
> **Response to reviewer FUtZ**
>
> Thank you for your helpful comments! We are glad that you find our paper to be novel and clear. Below are our responses to your questions:
>
> ### W1: I'm not sure how much the final sentences make sense… totally destroy any sort of meaning. Then again, in knowledge distillation, we might not care about this and only care about matching two functions point-by-point. I'm not sure how that's better than creating gibberish sentences as augmentation though.
>
> A: Thank you for pointing out an essential baseline for comparison! The reviewer raises an interesting question about the semantic meaning of augmented data. First, our motivation focuses on the perspective of machine learning, i.e., avoiding shifting decision boundaries by converting representations to tokens. From the semantic perspective, we agree with the reviewer that augmented sentences may have grammatical errors, "meaningless" tokens, and may be less meaningful than original sentences. However, "meaningless" to humans does not suggest meaningless to models, as AugPro indeed boosts the distillation performance. Besides, augmented sentences are not totally semantically meaningless to humans (Appendix C shows some examples). We understand the reviewer's concern about the performance of randomly generated data, as they can be generated easily but without any semantic meaning. To this end, we show the distillation results with randomly generated augmented sentences (i.e., randomly choosing tokens from the vocabulary and concatenating them to form a sentence) below. Gibberish sentences are not reasonable solutions as they do not bring performance gain compared to vanilla knowledge distillation. Results suggest that AugPro is helpful not because of the meaningful or meaningless semantics but because of avoiding shifting decision boundaries. We also add this discussion to Appendix C.
>
> |  Method    | SST-2 | CoLA | MNLI-M/MM | QNLI | QQP | MRPC | STS-B | RTE |
> |:-:|:-:|:-:|:-:|:-:|:-:|:-:|:-:|:-:|
> |       KD    | 92.09 | 45.56 | 85.93/85.61  | 89.46 | 91.36/88.24 | 84.56/88.85 | 87.29/87.18 | 61.37|
> |   MixUp  | 93.23 | 51.63 | 86.73/86.69  | 91.31 | 91.82/88.97 | 88.48/91.68 | 87.47/87.33 | 62.82 |
> |AugPro-Mix| 94.38 | 57.60 |  87.40/87.27 | 92.06 | 92.06/89.23 | 89.46/92.34| 88.10/87.87 | 71.84 |
> | **Random** | 92.78 | 45.89 |  86.37/85.55 | 89.35 | 91.42/88.43 | 84.07/88.93 | 87.72/87.57 | 61.37 |
>
> ### Q1:  I don't believe the code is available. The authors added implementation details in Appendix B, but that's probably not sufficient to reproduce the experiments.
> A: Please see our general response.

---

### Official Review · Reviewer_Lqpu · 2022-10-25

**Confidence:** 4
**Clarity, Quality, Novelty And Reproducibility:** The novelty is obvious, but the clari…
**Correctness:** 3
**Technical Novelty And Significance:** 3
**Empirical Novelty And Significance:** 3
**Recommendation:** 8

**Strength And Weaknesses:**

Pros:
1. The augmentation and projection method is rational intuitively and methodologically, knowledge distillation from large PLM is a good application scenario of the proposed method. Though the author did not clearly claim the motivation of projection from the perspective of linguistics, it seems to try to look for some keywords that can summarize the augmentation sample.
2. By combining augmentation and adversarial training together, the proposed method achieves good experimental results.
3. The effectiveness analysis is helpful in understanding the proposed method and good ablation study.

Cons:
1. There is one significant point that should be clearly explained. Equation 5 outputs a token list that can project embedding to real tokens. Technically, the token list should not be a sentence. But the examples from Appendix C are “real” sentences, though it is hard to read. This is amazing and beyond my expectation. How to convert a chaotic token list to a sentence? And there are a lot of stop words like “the” in the augmentation samples. If the project objective function aims to find keywords, I think the possibility of having stop words is really low. On the other hand, equation 5 takes the token that maximizes the similarity function from the vocabulary. But it seems there is a token list, which means all tokens in the selected token list have the distinctive similarity value. It would be helpful to explain equation 5 and its associated implementation in more detail. If more details are available, I would definitely raise my score.
2. The section 3 motivation samples are good, but I would suggest removing it since the difficulty in inverting an embedding to real tokens is familiar to most people, a short paragraph can be enough. More spaces should be provided for explaining the details of equation 5.
3. Better results are available if combing AugPro-FGSM and AugPro-Mix, but how to combine them together? This is because AugPro-MixUp and AugPro-FGSM may generate different token lists. Given the same objective function, the gradient derived from augmentation samples and adversarial samples are orthogonal or inverse if linearly combined them together like Loss_mix + Loss_fgsm? Provided with different input samples, it is not easy to determine the best optimization direction. Or it may be iterative training in real implementation.

**Summary Of The Paper:**

In the context of data augmentation for knowledge distillation, this paper proposes a method AugPro to address one significant and fundamental problem in representation learning for NLP: how to invert continuous representation to discrete tokens. AugPro finds tokens of which representations are close to the representation of an input augmentation sample, in the semantic space, determining what tokens can approximate the representation of augmentation samples acquired with MixUp. AugPro is validated with GLUE tasks and it shows good results in these tasks.

**Summary Of The Review:**

The idea from this paper is reasonable, but there is a huge gap between the idea and real implementation. The author should show much more detail about how to get augmentation samples from the selected token list.

---

> ### Author Response · Authors · 2022-11-16
> **Response to reviewer Lqpu**
>
> Thank you for your constructive feedback! We appreciate that you find our method novel and useful. We understand that you have confusion about our paper, here are our responses to your comments:
>
>
> ### Q1: Though the author did not clearly claim the motivation of projection from the perspective of linguistics, it seems to try to look for some keywords that can summarize the augmentation sample.
>
>
> A: The reviewer raises a fascinating question about the linguistic meaning of augmented data. In our paper, the motivation for projection mainly comes from the machine learning perspective, i.e., avoiding shifting decision boundaries. Linguistically speaking, projection may generate meaningful tokens (e.g., keywords that can summarize samples, as the reviewer mentioned) or “meaningless” tokens (they seem to be meaningless to humans but may be meaningful to models since they indeed boost the distillation performance). For example, we will get an augmented sentence “watch good video at home” when applying AugPro-Mix to samples “watch on video at home” and “as good”. The augmented sentence here is more like summarizing two samples, similar to what the reviewer points out. Some augmented sentences in Appendix C contain “meaningless” tokens to humans, but the whole sentences still maintain certain readability and semantic meanings. For example, the augmented sentence “River judgment Mit handy to consider the broader public interest” generated by applying AugPro-FGSM to the sentence “The judgments need to consider the broader public interest.” contains “meaningless” tokens. The usefulness of “meaningless” tokens also suggests that we should focus more on AugPro from the machine learning perspective, which is exactly our motivation. Again, we thank the reviewer for pointing out the missing discussion about the significance of linguistics. More discussion can be found in Appendix C in our updated PDF.
>
>
> ### W1.1: How to convert a chaotic token list to a sentence?
> A: Thanks for the question on the detailed implementation. In our work, we use T5 for experiments which uses SentencePiece (https://github.com/google/sentencepiece) as the vocabulary tool that supports the precise de-tokenization for arbitrary token lists.
> Specifically, for a given sentence, the tokenization step will replace white spaces “ ” with underscores “_” and split the sentence into subwords: e.g. “This is a sentence” -> [“This”, “_is”, “_a”, “_sent”, “ence”].
> The de-tokenization process the algorithm will do the reverse procedure: it will concatenate the tokens in sequence and replace underscores with white spaces. i.e. “”.join(tokens).replace(“_”, “ ”).
> Note: BERT uses WordPiece, which cannot guarantee precise de-tokenization. We added this detail to Appendix C to help readers understand the generation process.
>
>
> ### W1.2: And there are a lot of stop words like “the” in the augmentation samples. If the project objective function aims to find keywords, I think the possibility of having stop words is really low.
> A: We understand your concern about stopping words and keywords from the linguistic perspective. We hope our response to Q1 can solve your concerns. In short, generating tokens with linguistically concrete meaning is not our primary focus. Instead, we aim to reduce shifting decision boundaries by converting representations to tokens. We thank the reviewer for discussing the linguistic meaning; this will be one of our future works.
>
> ### W1.3: It would be helpful to explain equation 5 and its associated implementation in more detail.
> A: We are sorry for the confusion. Equation 5 is a token-wise nearest-neighbor algorithm, and the cosine similarity defines the distance. Explicitly speaking, let us assume $\mathbf{x}\_{rep}$ is one instance of $\mathcal{B}\_{rep}$, and $\mathbf{x}\_{rep} = [\mathbf{h}\_1, \cdots, \mathbf{h}\_n]$, where $\mathbf{h}\_i$ is the augmented embedding $\in \mathbb{R}^d$ and $d$ is the embedding dimension. The vocabulary $V$ has $|V|$ tokens, and each word $w\_i \in V$ has a corresponding embedding $\mathbf{e}\_i \in \mathbb{R}^d$. Then the pseudo-code of Equation 5 is as follows:
>
> \begin{equation}
>   \begin{aligned}
>      for\ \mathbf{h}\_i\ in&\ \mathcal{x}\_{rep}: \\\\
>     for\ &w\_j\ in\ V: \\\\
>     	&similarity\_{ij} = cosine(\mathbf{e}\_j, \mathbf{h}\_i) \\\\
>     w'\_{i} =  &argmax\_{w\_j}\ similarity_{ij}
>   \end{aligned}
> \end{equation}
>
> Where $w'\_{i}$ is a token $\in V$ and the output of AugPro for $\mathbf{h}\_i$.
>
> The augmentation sentence (as Appendix C shows) is the concatenation of AugPro tokens, i.e., $w'\_{i}$. In reality, the above codes can be simplified to a simple matrix multiplication, which can be found in Appendix B.2 in the updated PDF and in our general response.
>
> ### W2: Remove Section 3 (Or use only one paragraph) to leave spaces to illustrate details in Section 5.
> A: Thank you for your suggestion! We have added the implementation details of Equations 5 and 7 in Appendix B.2 in the updated PDF.

---

> > ### Author Response · Authors · 2022-11-16
> > **Response to reviewer Lqpu (Cont.)**
> >
> > ### W3: How to combine AugPro-Mix and AugPro-FGSM together?
> > A: We are sorry for missing the explanation and causing you confusion! The difference between AugPro methods is the selection of $\mathcal{L}\_{Aug}$. To combine the two methods, we only need to combine two losses. Specifically, the process is similar to Algorithm 1 and is as follows:
> > * For each batch B, we use MixUp or FGSM to get augmented representations $\mathcal{B}\_{MixUp}$ and $\mathcal{B}\_{FGSM}$.
> > * We apply Equations 5 and 7 to $\mathcal{B}\_{MixUp}$ and $\mathcal{B}\_{FGSM}$ to get $\mathcal{B}\_{AugPro-Mix}$ and $\mathcal{B}\_{AugPro-FGSM}$, respectively.
> > * The training loss will be $\mathcal{L} = \mathcal{L}\_{KD} + \mathcal{L}\_{AugPro-Mix}(\mathcal{B}\_{AugPro-Mix}) + \mathcal{L}\_{AugPro-FGSM}(\mathcal{B}\_{AugPro-FGSM})$
> > * Use SGD to update the student model $\theta\_S$
> >
> >
> > ### Q2: Clarity and reproducibility are not good. There is a huge gap between the idea and real implementation. The author should show much more detail about how to get augmentation samples from the selected token list
> > A: Please see our general response.

---

> > > ### Comment · Reviewer_Lqpu · 2022-11-24
> > > **Acknowledge of the rebuttal**
> > >
> > > I would like to thank the authors for the detailed clarification of my concerns. I will raise my score.

---

### Author Response · Authors · 2022-11-16
**General Response**

We thank all the reviewers for their time and insightful comments! We are glad that all reviewers find our paper novel and useful. We have updated our paper according to reviewers’ suggestions to improve the clarity and reproducibility:
* Made Algorithm 1 clearer.
* We added implementation details of Equations 5 and 7 to Appendix B (Algorithm 2). We believe our method could now be reproduced based on Algorithms 1 and 2, and provide our simple implementation based on the glove embedding https://anonymous.4open.science/r/AugPro-Mix-ICLR/AugPro-Mix.py. We will release full codes once the paper is published.
* We added linguistic analysis of generated examples in Appendix C.
* We illustrated the method to convert token lists to real sentences in Appendix C.

We highlight all the modifications with the blue color.

[Minor] We also fixed typos and paraphrased several sentences to make the reading smoother. We do not highlight these changes to make reviewers not distracted by these minor modifications.

---

### Decision · Program_Chairs · 2023-01-20

**Decision:**

Accept: poster

**Justification For Why Not Higher Score:**

The method is simple and not that novel, so I don't think it's enough as a spotlight.

**Justification For Why Not Lower Score:**

good paper

**Metareview: Summary, Strengths And Weaknesses:**

This paper proposes a simple and effective data augmentation technique for the knowledge distillation of NLP models. The proposed method secures the diversity of augmentation based on interpolation and converts the interpolated data back into tokens at the embedding level to minimize the shift of the decision boundary by the interpolation-based methods.

All four reviewers agreed that the idea is novel, and the experimental results are convincing and consistent. In the discussion, there was no disagreement in accepting the paper. However, I think it needs improvement in terms of clarity. In rebuttal, the authors explained kindly and the reviewers cleared up the misunderstanding, but it seems that more kind writing is needed for the reader. In particular, the examples and figures in Section 3 are somewhat confusing, regardless of the message the authors are trying to convey (for examples, when I first saw this example, it wasn't clear what the unseen data is in your setup. or how the label changes to blue after projection, etc.) The concept figure for the proposed method seems to be helpful in directly understanding the proposed method, so it must be included in the final version.

**Note From Pc:**

if the above contains the word "oral" or "spotlight" please see: "oral" presentation means -> notable-top-5% and "spotlight" means -> notable-top-25%. As stated in our emails, we are disassociating presentation type from AC recommendations